# Mechanisms Regulating Abnormal Circular RNA Biogenesis in Cancer

**DOI:** 10.3390/cancers13164185

**Published:** 2021-08-20

**Authors:** Ying Huang, Qubo Zhu

**Affiliations:** Xiangya School of Pharmaceutical Sciences, Central South University, Changsha 410013, China; huanghy@csu.edu.cn

**Keywords:** circular RNAs (circRNAs), biogenesis, trans-acting proteins, cis-regulatory elements

## Abstract

**Simple Summary:**

Circular RNAs (circRNAs) are circular RNA molecules without a 5′ cap and a 3′ poly(A) tail structure, which play an important role in tumor development, invasion and metastasis, etc. However, the mechanism of circRNA dysregulation in cancer remains unclear. Different from the classic splicing of linear RNA, circRNA is formed by back-splicing and is regulated by many cis-acting elements and trans-acting proteins. Exploring how the dysregulation of cis-regulatory elements and trans-acting proteins in tumor cells affects the biogenesis of circRNA, which in turn affects the development and prognosis of cancer, is of great significance for circRNA to become a cancer biomarker and therapeutic target.

**Abstract:**

Circular RNAs (circRNAs), which are a class of endogenous RNA with covalently closed loops, play important roles in epigenetic regulation of gene expression at both the transcriptional and post-transcriptional level. Accumulating evidence demonstrated that numerous circRNAs were abnormally expressed in tumors and their dysregulation was involved in the tumorigenesis and metastasis of cancer. Although the functional mechanisms of many circRNAs have been revealed, how circRNAs are dysregulated in cancer remains elusive. CircRNAs are generated by a “back-splicing” process, which is regulated by different cis-regulatory elements and trans-acting proteins. Therefore, how these cis and trans elements change during tumorigenesis and how they regulate the biogenesis of circRNAs in cancer are two questions that interest us. In this review, we summarized the pathways for the biogenesis of circRNAs; and then illustrated how circRNAs dysregulated in cancer by discussing the changes of cis-regulatory elements and trans-acting proteins that related to circRNA splicing and maturation in cancer.

## 1. Introduction

Circular RNAs (circRNAs) are a class of endogenous RNAs, which form a covalently closed continuous loop structure without 5′-3′ polarity nor polyA tail [1]. As an evolutionarily conserved RNA across phyla, circRNA plays essential roles in epigenetic regulation of gene expression at both the transcriptional and post-transcriptional level [2,3], such as acting as a ‘sponge’ through competitive binding of miRNA [4], forming RNA-protein complexes that regulate gene transcription [5,6], interacting with RNA-binding proteins as protein scaffolds [7], and even encoding proteins [8]. Growing evidence has confirmed that circRNAs involve in the tumorigenesis and metastasis of cancer [9]. Numerous circRNAs are considered to be promising biomarkers for diagnosis and prognosis in different cancers; and some of them are even developed as new therapeutic targets for cancer treatment [10,11]. Mechanism studies have also gradually revealed the important roles of circRNA in cancer, including cell proliferation and apoptosis, angiogenesis, epithelial to mesenchymal transition (EMT), tumor microenvironment, and drug resistance [9]. However, why circRNAs are abnormally expressed and how the circRNAs are regulated in tumors remain unknown.

Different from the classic canonical splicing of linear RNA, the biogenesis of circRNAs via back-splicing reversely connects the downstream (3′) splice donor site to an upstream (5′) splice acceptor site covalently [12]. Two hypotheses of back-splicing are highly recognized until now: the “lariat-driven circularization” and the “direct back-splicing”. The former model starts with canonical splicing for a linear RNA with skipped exons and a long lariat containing circRNA sequence, which is then further back-spliced to form a circRNA, while the latter model starts with back-splicing directly for a circRNA together with an exon-intron(s)-exon intermediate [13]. These two processes both require spliceosomal machinery, and are elaborately regulated by both *cis*-regulatory elements and *trans*-acting factors [14]. The *cis*-regulatory elements include the splice sites, enhancers, and silencers, especially the intronic complementary sequences (ICSs) near the junction points, such as the inverted *Alu* repeats [15]; and the *trans*-acting factors include spliceosome factors, cleavage factors RNA helicases, and RNA-binding proteins (RBP) [16]. Just like the alternative splicing (AS) and alternative polyadenylation [17] of linear RNA occur frequently with the disordered expression of splicing and maturation factors in cancer [18], aberrant biogenesis of circRNA is always observed in cancer with the dysregulation of spliceosomal machinery, and accumulating evidence indicates that the expression of circRNA is commonly altered in varies malignancies [19].

In this review, we summarized the pathways for the biogenesis of circRNAs; and then illustrated how circRNAs dysregulated in cancer by discussing the changes of cis-regulatory elements and trans-acting proteins that related to circRNA splicing and maturation in cancer.

## 2. The Biogenesis of circRNAs

The biogenesis of circRNAs via back-splicing is generally coupled with but different from the canonical splicing of linear RNAs [20]. Moreover, it is also distinct from the formation of other types of circular RNAs, such as the intron lariat escaped from debranching [21], the rRNA intermediates [22], and the direct ligation product of single strand RNA [23]. Most highly expressed circRNAs are composed of multiple internal exons of pre-mRNAs and contain few introns, indicating that back-splicing and canonical splicing are coupled [20]. There are also some circRNAs containing truncated exons [24]. However, the co-expression of circRNAs and the corresponding putative linear RNAs with exon exclusion is not detected, possibly due to the fast degradation of untranslatable linear RNAs [25]. Recent advances even showed that back-splicing and canonical splicing competed each other for the same transcripts [26].

Most scientists believe back-splicing and canonical splicing are coupled processes that are both required for the formation of circRNA. According to which step comes first, the processes of circRNA biogenesis can be classified into two groups. Once canonical splicing happens first, the RNA precursor will generate a linear RNA with skipped exons and a long lariat containing introns and exons, which is then back-spliced to generate a circRNA (Figure 1A). We called this process as “exon skipping” or “lariat-driven circularization” model. On the other hand, if the back-splicing happens first, the RNA precursor will generate a circRNA directly together with an intermediate containing both introns and exons, which is further processed to produce a linear RNA (Figure 1B). This process is referred as “direct back-splicing”. Current researches demonstrate that these two proposed models are both effective in vivo [13,21]. According to the difference of circularization mechanism, the “direct back-splicing” process can be further divided into two groups, “intron-pairing driven circularization” and “RBP mediated circularization”. The former one induces circularization due to proximity between the splice sites by pairing between complementary motifs within the introns flanking the exons [27]; while the latter one promotes the formation of the circle forming junction by bringing the circle-forming exons within proximity of each other through the interaction between RBPs bound with the intronic sequence motifs [26,28].

In summary, the circRNA biogenesis is a complex process that involved spliceosomal machinery, complementary intron pairing, RBP binding, and so on. How these elements regulate circRNA expression as well as how they change during tumorigenesis are of great interest.

## 3. Trans-Acting Proteins and RNAs

A variety of trans-acting proteins have been reported to be involved in the biogenesis of circRNAs. By regulating the splicing process of pre-mRNA, they affect the production and expression of circRNA at the post-transcriptional level, and further affect the proliferation and migration of tumor cells and the prognosis of cancer patients. According to the different mechanisms of trans-acting factors, we divide them into four categories: spliceosome factors, cleavage factors, RNA helicases, and RNA-binding proteins; and explain separately to illustrate the biogenesis process of circRNA and its influence on cancer.

### 3.1. Spliceosome Factors

The spliceosome is a large complex composed of five small nuclear RNAs (snRNAs), which combine with proteins to form particles, called small nuclear ribonucleoproteins (snRNPs). In eukaryotes, it is composed of U1, U2, U4, U5, and U6 snRNPs and a large number of proteins [29,30]. Both the pre-mRNA exons and their flanking introns contain special snRNP protein binding sites, and snRNP regulates the production of circRNA by specifical interaction with the binding sites [31]. Spliceosomes can catalyze the splicing events of most transcripts and remove the most common type of introns [32].

U2 auxiliary factor (U2AF) is a spliceosome factor and a non-snRNP protein, which is necessary for the binding of U2 snRNP to pre-mRNA branch sites [33]. U2AF2 acts as an oncogene in several cancers, such as glioma, primary non-small cell lung cancer, and melanoma [34,35,36]. In glioma, the expressions of U2AF2 and circRNA ARF1 (cARF1) are both up-regulated, and they are associated with decreased survival rates of patients. Mechanism studies show that U2AF2 promotes the proliferation, invasion, and angiogenesis of human brain microvascular endothelial cells (HBMECs) by up-regulating the expression of cARF1 in gliomas [37].

Studies have found that changes in the expression levels of many core components of the spliceosome have a profound impact on the level of circRNA production. For example, an important part of the spliceosome U2 snRNP (SF3B1 or SF3A1), snRNP-U1-70K, and snRNP-U1-C involve in 5′ splice site recognition, and the largest protein component of the spliceosome cross-links with U5 and U6 snRNAs (prp8), etc. When they are exhausted, the splicing products of pre-mRNA will shift to the direction of circRNA production, and the production of linear RNA will decrease [16]. Among them, the mutation of SF3B1 is considered to be a common driving factor of hematological malignancies [38]. Mutations in SF3B1 are found in most patients with myelodysplastic syndrome (MDS) [39,40]. The mutation of SF3B1 has also become an important factor in a variety of pigmented tumors, such as uveal melanoma (UM) [41], leptomeningeal melanoma [42], and blue nevus-like cutaneous melanoma [43]. In MDS and UM, compared with SF3B1 wild-type patients, patients with SF3B1 mutations have a higher 5-year survival rate and a better cancer prognosis [40,41]. However, there is no direct evidence that the spliceosomes affect the procession of cancer by affecting the production of circRNA, and it will be the direction of continued research and exploration in the future.

### 3.2. Cleavage Factors

According to the reports, several cleavage factors can directly cut the adjacent intron sequences of exons forming circRNA to promote the formation of circRNA [28].

For example, the cleavage factor ESRP1 can evenly target circRNA GGT-rich sequences to accelerate the circRNA biogenesis, such as circANKS1B and circUHRF1. Clinically, in breast cancer, the expression of ESRP1 is positively correlated with circANKS1B. The high expression of circANKS1B is closely related to the clinical stage and is an independent risk factor for the overall survival rate of breast cancer patients. At the same time, breast cancer patients with high expression of ESRP1 also have a poor overall survival [44]. In oral squamous cell carcinoma (OSCC), circUHRF1 is significantly up-regulated. The overexpression of circUHRF1 is closely related to the poor prognosis of OSCC patients and promotes tumor proliferation, migration, invasion, and epithelial mesenchymal transformation (EMT) [45].

The cleavage factor NUDT21 is a part of the cleavage factor Im (CFIm) complex. In the form of a dimer, it combines with two UGUA sequences upstream of poly(A) to form a loop to prevent the splicing of pre-mRNA by the CPSF family, thereby promoting the formation of circRNA [46]. In hepatocellular carcinoma (HCC), the expression level of circRNA is generally down-regulated, and the level of NUDT21 is also significantly down-regulated, HCC patients with low levels of NUDT21 have poor overall survival [47]. In short, the down-regulation of NUDT21 in HCC inhibits the production of circRNA containing UGUA sequence, which in turn leads to the imbalance of the proliferation of HCC cells [48].

As the core component of the 3′-end splicing complex, Cleavage and Polyadenylation Specific Factor 4 (CPSF4) also has a regulatory effect on the formation of circular RNA. It can specifically bind to the PAS regulatory elements on the precursor mRNA, and mediates the cleavage of the CPSF family complex, thereby inhibiting the formation of circular RNA and affecting the expression of the formed circular RNA. The high expression of CPSF4 in HCC also corresponds to the overall down-regulation of circular RNA. Further studies demonstrated that overexpression of CPSF4 could promote the proliferation of HCC cells and was closely related to the clinicopathological characteristics of HCC patients [49].

### 3.3. RNA Helicases

RNA helicases, a family of proteins that promote the unwinding of RNA during splicing and translation, play important roles in circRNA biogenesis because unwinding of the paired RNA sequence is a critical process for RNA circulation.

DExH-box helicase 9 (DHX9) is a nuclear RNA helicase that can interact with inverted complementary sequences and unwind RNA pairs flanking circularized exons to prevent the production of circRNAs [50]. In lung adenocarcinoma (LUAD), overexpressed CircDCUN1D4 significantly inhibits tumor metastasis, reduces glycolysis, and can be used as an independent prognostic factor for LUAD patients. Researchers found that DHX9 interacted with AluJo/AluSc binding sites to reduce the expression of circDCUN1D4. The expression of DHX9 was negatively correlated with the expression of circDCUN1D4 in LUAD tissues, and the lower circDCUN1D4 level in LUAD was due to the up-regulation of DHX9 [51]. In addition, studies have found that DHX9 can negatively regulate circPICALM levels by binding to its inverted repeats Alu (IRAlus). CircPICALM was downregulated in bladder cancer, and low circPICALM expression was related to advanced T stage, high grade, and poor overall survival rate [52]. Unlike circDCUN1D4 and circPICALM, DHX9 post-transcriptionally regulated the biogenesis of Circular RNA cSMARCA5 by binding to other nonrepetitive but inverted complementary sequences instead of inverted repeated Alu pairs, such as I14RC (reverse complementary sequences in intron 14) and I16RC (reverse complementary sequences in intron 16). DHX9 was significantly upregulated in HCC and the expression of cSMARCA5 was negatively correlated with the histochemical score of DHX9 in 40 HCC tissues. Furthermore, the tumor suppressor cSMARCA5 plays an important role in the growth and metastasis of HCC and can be used as a biomarker for the prognosis of patients undergoing liver cancer resection [53].

Eukaryotic initiation factor 4A3 (EIF4A3, DDX48) is a member of the DEAD box family, which is one of the largest helicase families characterized by an Asp-Glu-Ala-Asp (DEAD) motif [54]. EIF4A3 does not show helicase activity by itself [55]; only when the other 2 core components of exon junction complex (EJC) MLN51 and Magoh/Y14 bind with EIF4A3 together, the ATPase and helicase activities of EIF4A3 are stimulated [56], and the pre-mRNA splicing process starts. The expressions of circMMP9 and circASAP1 are significantly upregulated in glioblastoma multiforme (GBM), and they promote the proliferation, migration, and invasion of GMB. Through the circinteractome database, the author found that EIF4A3 had four binding sites in the upstream region of circMMP9 mRNA transcript, which could promote the circularization of exons 12 and 13, and then promoted the expression of circMMP9 [57]. Similar to circMMP9, EIF4A3 is also positively correlated with circASAP1. However, EIF4A3 increases the expression of circASAP1 by binding with the downstream flanking sequence of circASAP1 instead of the upstream sequence [58]. In triple-negative breast cancer (TNBC), circSEPT9 is significantly up-regulated, and is closely related to the clinical stage and poor prognosis of TNBC patients. Recent research has demonstrated that EIF4A3 increased the expression of circSEPT9 by binding to the SEPT9 pre-mRNA, and modulated cell cycle [59].

ADAR1 is an RNA editing enzyme that can catalyze adenosine (A) in the reverse complementation region of introns to inosine (I), untie the pairing structure, and prevent the splicing site from approaching for reverse splicing, thereby strongly inhibiting the generation of circRNA [60,61]. Studies have shown that ADAR1 is closely related to the poor prognosis of liver cancer. ADAR1 inhibits the production of CircARSP91 in HCC and further promotes the proliferation of HCC [61]. ADAR1 overexpression is also one of the reasons for the down-regulation of hsa_circ_0004872 expression in gastric cancer tissues. Has_circ_0004872 acts as a “miRNA sponge” to inhibit the proliferation, invasion, and migration of gastric cancer cells, and can serve as a diagnostic and prognostic marker for gastric cancer patients [62]. In pancreatic ductal adenocarcinoma (PDAC), circNEIL3 plays a role of oncogene and promotes the progression and metastasis of PDAC. ADAR1 can inhibit the formation of circNEIL3 by catalyzing the A of NEIL3 complementary sequence to G and affect the clinical stage and survival of PDAC patients [63].

### 3.4. RNA-Binding Proteins

Except for the spliceosome factors, cleavage factors, and RNA helicases, several RNA-Binding Proteins (RBPs) can shorten the distance between the receptor site and the donor site through interaction with the flanking introns of exons, which promotes the production of corresponding circRNAs [64], such as heterogeneous nuclear ribonucleoprotein L (HNRNPL), QKI, FUS, etc.

HNRNPL can bind to the flanking introns of exons to promote the production of circRNA, especially when it exists on both sides [65]. In Gastric cancer [50], the expression of circLMO7 is significantly up-regulated. TCGA database shows that HNRNPL is highly expressed in gastric cancer tissues and is positively correlated with the expression level of circLMO7. Deep exploration shows HNRNPL promotes the production of circLMO7 by interacting with the four binding sites of the flanking introns of the circLMO7 exons [66].

The Quaking (QKI) belongs to the evolutionarily conserved signal transduction and activator of RNA [67] family and contains a single heterogeneous nuclear ribonucleoprotein K homology (KH) RNA-binding domain [68]. QKI dimers can bind with two separate RNA recognition elements on a single RNA molecule [69]. Therefore, QKI may promote the biogenesis of circRNA by binding with pre-mRNA to bring circ-forming exons closer to each other [28]. CircZKSCAN, which is an independent and significant factor for HCC, is positively correlated with the survival rate of patients; and the expression of QKI is positively correlated with the expression of circZKSCAN. The abnormal down-regulation of QKI in HCC tissue may lead to the down-regulation of circZKSCAN1 and accelerate the growth of cancer cells [70].

FUS, a well-known proto-oncogene, is reported to regulate the expression of circRNA in murine embryonic stem cell-derived motor neurons [17]. In primary colorectal cancer tissues, the expression of CircLONP2 is significantly up-regulated, and the expression of CircLONP2 is positively correlated with the mRNA level of FUS. High expression of FUS is responsible for the low overall survival rate of patients with colorectal cancer, and poor clinicopathological characteristics. Studies have shown that circ-LONP2 is the first circRNA identified as an accessory component of the microprocessor complex, expanding the research on potential regulatory functions of circRNAs in both physiological and pathological processes [71]. Furthermore, the reverse complementary matches in intron 7 of CircLONP2 are composed of repeated “GUUG” or “ACUU” regions, which are just between the binding motifs of FUS. Mechanism studies show that FUS regulates the metastasis of colorectal cancer cells by interacting with the reverse complementary matches of circLONP2 to regulate the biogenesis of circLONP2 [72]. In lung cancer, CircZNF609, which functions as an onco-circRNA, is highly expressed, and promotes the proliferation and migration of cancer cells. FUS RNA-binding protein can bind to the upstream of ZNF609 pre-mRNA to induce the formation of CircZNF609 and increase its level in lung cancer [73]. In TNBC, circHIF1A is significantly overexpressed and is associated with metastasis, poor prognosis and TNBC subtypes. The RNA-binding protein FUS regulates the biogenesis of circHIF1A by interacting with flanking introns, and significantly promotes TNBC growth and metastasis [74].

## 4. Cis-Regulatory Elements

In addition to trans-acting factors that regulate circRNA biogenesis, many cis-regulatory elements can also influence the cyclization of RNA. In particular, reverse complementary sequences (including repeated complementary sequences, such as the Alu element and non-repeated complementary sequences) and direct-action sites of some splicing factors (such as the PAS site, UGUA, and GGT-rich) play important roles in the generation of circRNA.

### 4.1. Intronic Complementary Sequences

Bioinformatics analysis shows that there are long introns at both ends of the reverse spliced circRNA exons. According to the calculation, the presence of Alu in the flanking introns is highly correlated with the formation of circRNA [27]. The pairing of the inverted repeat sequence of the Alu element makes the downstream splice donor and upstream splice acceptor sites close to each other, thereby promoting the occurrence of reverse splicing events [27]. Alu elements are inverted repeats on the introns flanking exons, which can play a key role in the formation of circRNA together with trans-acting proteins. For example, DHX9 can combine with the Alu inverted repeats on circDCUN1D4 [51] and CircPICALM [52] to inhibit the generation of circRNA, thereby inhibiting the proliferation, metastasis and invasion of lung cancer and colon cancer respectively, which is of great significance to the prognosis of cancer patients.

The formation of circRNA depends on the pairing ability of complementary sequences, which can be repetitive sequences such as Alu elements, or non-repetitive but complementary sequences. For example, intron 14 and intron 17 of CircLONP2 have a highly matched sequence. Intron 17 is composed of several repeated “GUUG” or “ACUU” regions, which are located exactly within the binding motif between the proto-oncogene FUS. FUS binds to the flanking introns of CircLONP2 to promote its generation [72]. In addition, colorectal cancer cells with up-regulated expression of circLONP2 have significantly enhanced migration and invasion capabilities. DHX9 can also bind to the reverse complementary sequence flanking the circSMARCA5 intron to reduce its expression, which is of great significance for inhibiting the proliferation and metastasis of HCC [53]. Moreover, the longer flanking introns are not necessary for circular RNA formation, but the extended length can introduce more Alu elements that, in turn, promote exon circularization. [20]. The distance and competition between the complementary sequences in the flanking of introns will affect alternative splicing, thereby affecting circRNA circularization, and may also lead to the occurrence of alternative reverse splicing at the same locus, resulting in a variety of circRNAs [75].

But not all intron complementary sequences can promote pre-mRNA reverse splicing and exon circularization, the hairpin structure caused by base pairing may inhibit the biogenesis of circRNA [76]. For example, the 321 to 343 nucleotides of EPHB4 locus have a low-complexity sequence of poly(A). The binding of the Poly(A) binding protein (PABP) to this region can spatially interfere with its hairpin formation ability, thereby promoting the formation of EPHB4 circRNA. In addition, studies have shown that circEphB4 is up-regulated in glioma cells and promotes the proliferation and glycolysis of glioma cells, which is closely related to the poor prognosis of patients [77].

### 4.2. Splice Sites

Both the RBPs and the mutual pairing sequences regulate circRNA generation by bringing upstream splicing receptors and downstream splicing donors close to each other. While the splicing factor can directly cleave the splice site on the intron to regulate the biogenesis of circRNA.

The polyadenylation site (PAS), which plays an important role in the process of pre-mRNA 3’-UTR cleavage and maturation, plays an important role in regulating the expression of circular RNA. Among them, AAUAAA is the classic PAS motif [78], and other variants include AUAAAA, AUUAAA, CAAUAA, AAAAAA, AUAAAG, etc [79]. The PAS site is usually located 10–30 nucleotides upstream of the splice site. The PAS element can be recognized by the cleavage factors to promote the cleavage of the circRNA and reduce its expression. Recent studies have shown that CPSF4 can inhibit the formation of circRNA by recognizing and cutting the AAUAAA motif on pre-mRNA, thereby inhibiting the proliferation of HCC cells, which is of great significance to the prognosis of HCC patients [49].

The two UGUA elements located upstream of the PAS site can also regulate RNA cyclization. In HCC, the UGUA sequence located at the proximal end of the 3’-UTR is considered to be the splicing and polyadenylation site in the alternative splicing and alternative polyadenylation [17] process, which can bind to the RNA splicing complex protein to regulate the length of the 3’-UTR; and then regulate the APA process [80]. Recent studies have shown that the UGUA sequence in the pre-mRNA in HCC is an important cis-regulatory element in the production of circRNA. It can be recognized by the cleavage complex and combine with it to form a circular structure, thereby it avoids the shearing of pre-mRNA by the cleavage protein and promotes the looping process of circular RNA [48]. For example, in HCC tissues, NUDT21 combines with the two “UGUA” sequences upstream of poly(A) in the form of dimers to form a loop, which prevents the cleavage factor from pre-mRNA cleavage and promotes the production of circRNA. Finally, overexpression of NUDT21 leads to the proliferation inhibition of HCC cells [46,48].

In addition, studies have shown that the GGT-rich sequence on the flanking intron of the pre-mRNA exon is the binding sequence of the splicing factor ESRP1. ESRP1 is closely related to the poor prognosis of cancer patients, because ESRP1 can directly bind to the GGT-rich sequence to cut the circRNA, thereby accelerating the circRNA (such as circUHRF1 and circANKS1B) biogenesis, which in turn promotes tumor growth and metastasis [44,45].

## 5. Conclusions

circRNA has been proven to be an important biomarker of cancer in many studies. In this review, starting from the biogenesis mechanism of circRNA, we further explored the causes of circRNA dysregulation in cancer from two aspects of trans-acting factors and cis-acting elements, as well as the relationship with the occurrence, development, and prognosis of cancer (Table 1). There are different ways to affect the circularization of circRNA. For the trans-acting elements, spliceosome binds to the binding site and facilitates the pre-mRNA splicing (Figure 2A), cleavage factors cut the pre-mRNA directly (Figure 2B), RNA helicases unwrap the paired complementary sequences (Figure 2C), and RBPs make the intron complementary sequences close to each other (Figure 2D). Among the cis-acting elements, the pairing of intron complementary sequences (Alu elements) and the binding of splice sites (PAS, UGUA, and GGT-rich) are essential for the formation of circRNA. Mutations of trans-acting factors and cis-acting elements may affect the expression of circRNA, which in turn affects the progression of a variety of cancers. However, the current research on the formation mechanism of circRNA and its potential relationship with cancer is still in its infancy [81]. It is believed that with the continuous in-depth study of circRNA regulation mechanism, circRNA is not only a biomarker, but may also become a potential therapeutic target for cancer.

## Figures and Tables

**Figure 1 cancers-13-04185-f001:**
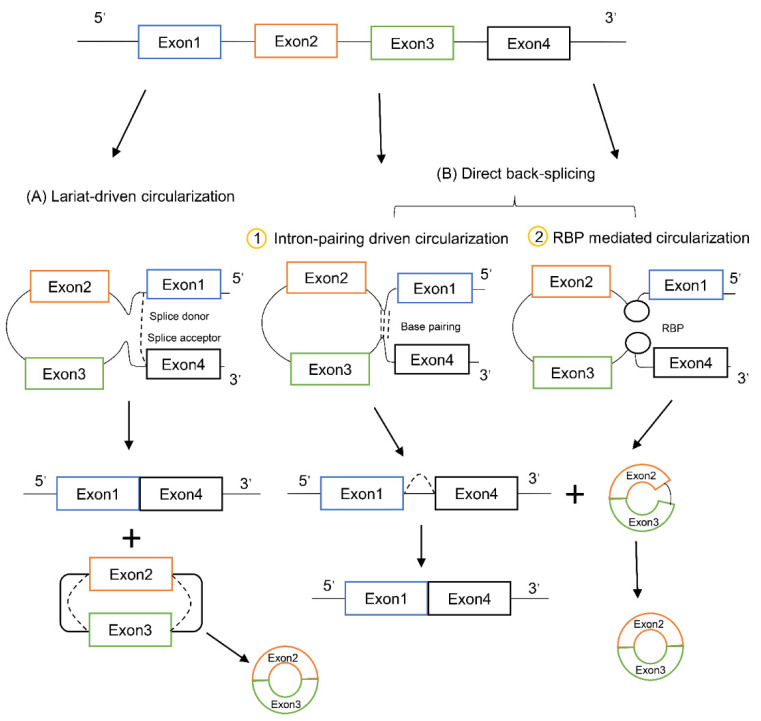
Biogenesis of circRNA. (**A**) Canonical splicing occurs first, exon skipping forms an mRNA composed of exons 1 and 4, and a lariat structure composed of exons 2 and 3, and then back-splicing produces circRNA. (**B**) Back-splicing occurs first, circRNA containing exons and introns is generated through back-splicing, and then further processed to generate linear mRNA. It can be divided into two types, one is (1) “intron-pairing driven circularization”, the complementary sequences of the flanking introns pair with each other to bring the splice sites closer to promote circularization, and the other is (2) “RBP mediated circularization”, RBP interacts with the intron binding sites to make the splice sites closer to induce circularization.

**Figure 2 cancers-13-04185-f002:**
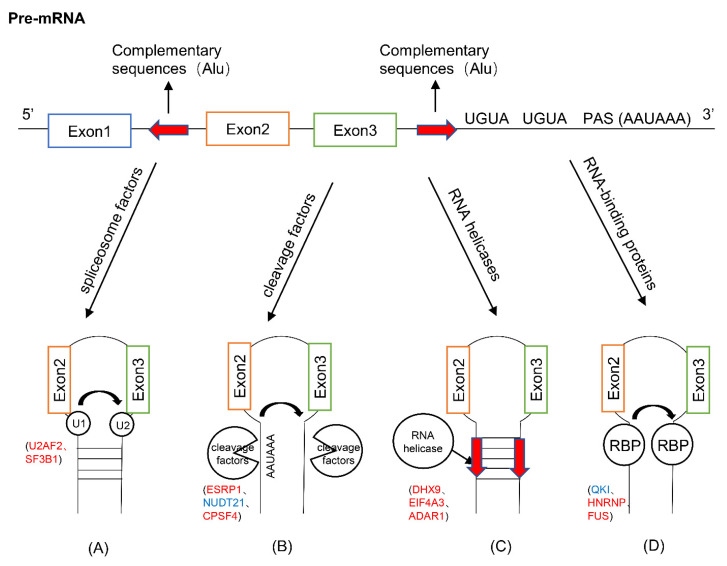
The roles of cis-acting elements and trans-acting factors in circRNA biogenesis. (**A**) The combination of spliceosome factors (U2AF2, SF3B1) with pre-mRNA binding sites. (**B**) Cleavage factors (ESRP1, NUDT21, CPSF4) directly cleave the cis-acting elements (PAS, UGUA) on pre-mRNA. (**C**) RNA helicases (DHX9, EIF4A3, ADAR1) unwind or interfere with the intron complementation Sequence (Alu). (**D**) RNA-binding proteins (QKI, HNRNP, FUS) bind to the pre-mRNA binding site to make the splice sites close. For the trans-acting factors, the ones labeled in red indicate increased expression in cancer, and the blue ones indicate decreased expression in cancer.

**Table 1 cancers-13-04185-t001:** Trans-acting proteins and RNAs for RNA circulation.

Trans-Acting Proteins	Proteins/RNA	Target circRNAs	Correlation	Cancer	Ref.
spliceosome factors	U2AF	cARF1	Positive	Glioma	[37]
cleavage factors	ESRP1	circANKS1B	Positive	Breast cancer	[44]
		circUHRF1	Positive	Oral squamous cell carcinoma	[45]
	NUDT21	circRNAs with UGUA	Positive	Hepatocellular carcinoma	[48]
	CPSF4	circRNAs with PAS	Negative	Hepatocellular carcinoma	[49]
RNA helicases	DHX9	circDCUN1D4	Negative	Lung adenocarcinoma	[51]
		circPICALM	Negative	Bladder cancer	[52]
		cSMARCA5	Negative	Hepatocellular carcinoma	[53]
	EIF4A3	circMMP9	Positive	Glioblastoma multiforme	[57]
		circASAP1	Positive	Glioblastoma multiforme	[58]
		circSEPT9	Positive	Triple-negative breast cancer	[59]
	ADAR1	CircARSP91	Negative	Hepatocellular carcinoma	[61]
		hsa_circ_0004872	Negative	gastric cancer	[62]
		circNEIL3	Negative	pancreatic ductal adenocarcinoma	[63]
RNA-binding proteins	HNRNPL	circLMO7	Positive	Gastric cancer	[66]
	QKI	circZKSCAN	Positive	Hepatocellular carcinoma	[70]
	FUS	CircLONP2	Positive	Primary colorectal cancer	[72]
		CircZNF609	Positive	Lung cancer	[73]
		circHIF1A	Positive	Triple-negative breast cancer	[74]

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
