# Peer review of "Mechanisms Regulating Abnormal Circular RNA Biogenesis in Cancer"

_cancers, 2021, doi:10.3390/cancers13164185_

Round 1

Reviewer 1 Report

This review by Ying et al aimed to summarize the deregulation of circRNA biogenesis in cancer. The manuscript is sound, I only have some minor comments for this work.

  • circRNAs are not a non-coding RNA category, since many of them encode peptides. The authors need to correct this.
  • Karousi et al (Int. J. Mol. Sci. 202021, 8867. https://doi.org/10.3390/ijms21228867) proposed some circRNAs that include truncated exons in CRC. The authors may refer to this or other related literature.
  • The manuscript needs slight grammar and syntax improvement.

Author Response

Response to Reviewer 1 Comments

Point 1: circRNAs are not a non-coding RNA category, since many of them encode peptides. The authors need to correct this.

Response 1: Thank you to point this out, the statement that circRNA is non-coding RNA in line 9, line 17 and line 32 has been deleted.

Point 2: Karousi et al (Int. J. Mol. Sci. 2020, 21, 8867. https://doi.org/10.3390/ijms21228867) proposed some circRNAs that include truncated exons in CRC. The authors may refer to this or other related literature.

Response 2: Thank you for your suggestion. We have cited this reference in line 76 of the review.

Point 3: The manuscript needs slight grammar and syntax improvement.

Response 3: Thank you for your suggestion. We have revised the grammar in many places in the review.

Reviewer 2 Report

This review of Ying et al. is a very good one, clearly explaining how the dysregulation of cis-regulatory elements and trans-acting proteins in cancer cells affects the biogenesis of circRNAs, thus affecting the development of cancer and prognosis of cancer patients. The authors have very efficiently summarized the current literature and provide their own critical point-of-view. I have only two minor comments to make, regarding the literature:

  • Lines 261-269: The authors refer to circ-LONP2, which is a prominent example. They should add the role of this circRNA along with an appropriate reference; importantly, as stated by Artemaki et al. [Cancers (Basel). 2020 Aug 31;12(9):2464. doi: 10.3390/cancers12092464.]: “circ-LONP2 is the first circRNA identified as an accessory component of the microprocessor complex, expanding the research on potential regulatory functions of circRNAs in both physiological and pathological processes“.
  • Lines 368-371: The authors should add a reference here; an absolutely relevant review to cite here is: “The role of circular RNAs in therapy resistance of patients with solid tumors” by Papatsirou et al. [Per Med. 2020 Nov;17(6):469-490. doi: 10.2217/pme-2020-0103].

Author Response

Response to Reviewer 2 Comments

Point 1: Lines 261-269: The authors refer to circ-LONP2, which is a prominent example. They should add the role of this circRNA along with an appropriate reference; importantly, as stated by Artemaki et al. [Cancers (Basel). 2020 Aug 31;12(9):2464. doi: 10.3390/cancers12092464.]: “circ-LONP2 is the first circRNA identified as an accessory component of the microprocessor complex, expanding the research on potential regulatory functions of circRNAs in both physiological and pathological processes”.

Response 1: Thank you for your suggestion. We have added the role of circLONP2 in lines 265-268 and cited this reference.

Point 2: Lines 368-371: The authors should add a reference here; an absolutely relevant review to cite here is: “The role of circular RNAs in therapy resistance of patients with solid tumors” by Papatsirou et al. [Per Med. 2020 Nov;17(6):469-490. doi: 10.2217/pme-2020-0103].

Response 2: Thank you for your suggestion. We have cited this reference in line 374 of the review.
